# Antiferromagnet–Ferromagnet Transition in Fe_1−x_Cu_x_NbO_4_

**DOI:** 10.3390/ma15217424

**Published:** 2022-10-22

**Authors:** Diego S. Evaristo, Raí F. Jucá, João M. Soares, Rodolfo B. Silva, Gilberto D. Saraiva, Robert S. Matos, Nilson S. Ferreira, Marco Salerno, Marcelo A. Macêdo

**Affiliations:** 1Department of Physics, Federal University of Sergipe, São Cristóvão 49100-000, Brazil; 2Department of Physics, State University of Rio Grande do Norte, Mossoró 59610-210, Brazil; 3Faculty of Education Sciences and Letters of Sertão Central, State University of Ceará, Quixadá 63900-000, Brazil; 4Postgraduate Program in Materials Science and Engineering (P2CEM), Federal University of Sergipe, São Cristovão 49100-000, Brazil; 5Chair of Materials Science and Technology, Technische Universität Dresden, Budapester Str. 27, 010169 Dresden, Germany

**Keywords:** structure properties, magnetic transition, antiferromagnetic–ferromagnetic

## Abstract

Iron niobates, pure and substituted with copper (Fe_1−x_Cu_x_NbO_4_ with x = 0–0.15), were prepared by the solid-state method and characterized by X-ray diffraction, Raman spectroscopy, and magnetic measurements. The results of the structural characterizations revealed the high solubility of Cu ions in the structure and better structural stability compared to the pure sample. The analysis of the magnetic properties showed that the antiferromagnetic–ferromagnetic transition was caused by the insertion of Cu^2+^ ions into the FeNbO_4_ structure. The pure FeNbO_4_ structure presented an antiferromagnetic ordering state, with a Néel temperature of approximately 36.81K. The increase in substitution promoted a change in the magnetic ordering, with the state passing to a weak ferromagnetic order with a transition temperature (T_c_) higher than the ambient temperature. The origin of the ferromagnetic ordering could be attributed to the increase in super-exchange interactions between Fe/Cu ions in the Cu^2+^-O-Fe^3+^ chains and the formation of bound magnetic polarons in the oxygen vacancies.

## 1. Introduction

FeNbO_4_ is a polymorphic compound that crystallizes in three different crystalline phases depending on the annealing temperature [1].Considered an n-type semiconductor, it has a narrow bandgap of 1.81–2.25 eV [2,3], making it attractive for multipleapplications, e.g., as a photocatalyst [4,5], due to its excellent visible light activity attributed to the higher energy levels of the Nb 4d orbital;in gas sensors [3], capacitors [6], and lithium-ion batteries [7,8]; and as an anode material in solid oxide fuel cells (SOFCs) [9].The most stable phase, under ambient conditions, has monoclinic symmetry (m-FeNbO_4_, space group P2/c) and is obtained at temperatures below 1085 °C [10]. Its structure has ordered cations, with both Fe^3+^ and Nb^5+^ forming regular octahedra, coordinated by six oxygen ions, forming zig-zag chains of FeO_6_ and NbO_6_ (see Figure 1a) [10,11]. A crystal structure with orthorhombic symmetry (o-FeNbO_4_, Pbcn space group) is formed in the temperature range of 1085 to 1380 °C [12].Unlike the m-FeNbO_4_ phase, above 1100 °C, the distribution of cations in this structure becomes disordered. Above 1380 °C, it crystallizes in the tetragonal phase (t-FeNbO_4_, space group P42/mnm) [12,13].

The order/disorder of the cations and the super-exchange interactions via oxygen are significant for the magnetic properties of these compounds, and the degree of the order depends on the specific conditions of synthesis [14]. m-FeNbO_4_ is isostructural to FeWO_4_, with the sharing of the edges of the Fe-O-Fe octahedral chains resulting in magnetic ordering, forming infinite ferromagnetic (FM) sheets along the [1 0 0] direction [1]. Cross-chains of Fe-O-Nb-O-Fe result in antiferromagnetic (AFM) arrangements and, therefore, in net antiferromagnetism [15]. Recently, Wang et al., through theoretical calculations of functional density (DFT), hypothesized the existence of three distinct and stable magnetic ordering configurations, which can be explained as the possible formation of magnetic domains in the material [10].

The o-FeNbO_4_ phase presents different magnetic behavior to the monoclinic phase. Cationic disorder disrupts the super-exchange, intra-chain, and inter-chain pathways, resulting in the absence of any magnetic ordering up to 4.2 K [1]. Lakshminarasimhan et al. compared the magnetic properties of both phases (m- and o-FeNbO_4_) and observed that cationic disorder in the orthorhombic structure could lead to magnetic frustration and thus induce spin-glass behavior, as well as spin-glass characteristics, such as memory effects [15].

The antiferromagnetic–ferromagnetic transition is a phenomenon that has been reported for several structures, such as FeRh [16], CeFe_2_ [17], ZnO [18,19], LaFeO_3_ perovskite-like structures [20], LaMnO_3_ [21,22], LaMn_0.5_Ni_0.5_O_3_ [23], YTiO_3_ [24], and LaCoO_3_ [25]. Several mechanisms have been suggested to explain the nature of the ferromagnetic state of these compounds: super-exchange, based on localized electronic interactions through an oxygen ion; double exchange via charge transfer; and itinerant electronic ferromagnetism [25].

Replacement at the lattice site by ions with different ionic radii is an effective way to modulate the crystal structure and magnetic ordering, which results in a reduction in the AFM and an increase in the FM of La_1−x_A_x_FeO_3_ (A = Bi, Al, Zn, Ce, etc.) [20]. Another way to modulate the magnetic ordering is substitution with non-magnetic acceptor ions, whereby the magnetic transition arises from super-exchange interactions between mixed-valence magnetic ions, as reported for the compound La_1−x_Sr_x_Mn_0.5_Ni_0.5_O_3_ [23]. Oxygen-mediated super-exchange interactions also play an essential role in magnetic ordering properties. Oxygen deficiency can change the magnetic ordering in structures such as LaCoO_3_ [25] and LaMnO_3_ [22]. One of the most efficient methods for changing the magnetic ordering is magnetic ion doping. Co-doping with the magnetic ions Cu and Fe is reported to be the main cause of the emergence of ferromagnetism in ZnO [18,19]. 

Ferromagnetic semiconductor materials have attracted much interest due to their potential application as spin-polarized carriers and their easy integration in semiconductor devices [19].

In this work, we evaluated the influence of the substitution of Fe^3+^ ions with Cu^2+^ ions on the structural and magnetic properties of ordered-structure FeNbO_4_. Our results showed that copper insertion induced an antiferromagnetic–ferromagnetic transition, which may have been associated with the increase in ferromagnetic interactions caused by the rise in super/exchange interactions between Cu^2+^-O-Fe^3+^ ions and the contribution of defects such as oxygen vacancies.

## 2. Materials and Methods

Fe_1−x_Cu_x_NbO_4_ samples (x = 0–0.15) were prepared by the modified solid-state reaction method. The compounds iron III nitrate (Fe(NO_3_)_3_∙9H_2_O), copper oxide (CuO), and niobium pentoxide (Nb_2_O_5_) were calculated stoichiometrically and dissolved in ethanol. The product was dried at 80 °C for 1 h in a mini-muffle furnace and sintered at 1050 °C for 16 h in an ambient atmosphere.

Structural characterization was carried out by X-ray diffraction (XRD) performed on a Miniflex instrument (Rigaku, Tokyo, Japan) using Cu-Kα radiation (λ = 1.54 Å) in the range of 10–80° with a 0.01° step and a counting time step of 2 s. The lattice parameters of the crystal structures were refined using the Rietveld method with GSAS software. 

Raman spectroscopy was carried out with a T64000 spectrometer (HORIBA-Jobin Yvon, Kyoto, Japan) using a 100 nW laser (λ = 514.5 nm) as an excitation source, with 2 cm^−1^ of spectral resolution. The temperature-dependent magnetization (field-cooled (FC) and zero-field-cooled (ZFC)) and magnetization (M) versus applied magnetic field (H) curves were obtained at a low temperature and ambient temperature using a physical property measurement system (PPMS) (Quantum Design, Les Ulis, France).

## 3. Results and Discussion

### 3.1. Structural Analysis

The samples were first characterized by XRD at room temperature. The resulting patterns are shown in Figure 1b. The analysis revealed that the samples presented a monoclinic phase, belonging to the P2/c space group (No. 13), indexed through the ICDD standard (00-070-2275), with the reflection planes (001) and (110) observed at 19.30° and 23.6° being characteristic of this type of structure [15]. Impurities were also observed, which were indexed with the following phases: Nb_2_O_5_ (ICSD-017027), NbO_2_ (ISCD-035181), and Fe_2_O_3_ (ICSD-082903). In Figure 1c,d, the patterns fitted by Rietveld refinement to x = 0 and 0.15 are presented, respectively. 

The values obtained for the refinement quality factors R_wp_, R_p_, and χ^2^ indicated that the refined results were reliable. The results obtained for all samples are listed in Table 1. The insertion of Cu^2+^ ions did not promote significant changes in the lattice parameters of the structure (see Table 1). However, it did cause local changes in the octahedral sites FeO_6_ and NbO_6_ (see Figure 1a) regarding the bond length and angle at each site.

The term χ^2^ in Table 1 indicates that this bond was repeated in the octahedron (see Figure 2b). A reduction in the Nb-O bond length was observed, while the Fe-O bond length increased because of the substitution. This was expected due to the small difference in the ionic radius of the substituent (Fe^3+^ = 0.064 nm and Cu^2+^ = 0.073 nm), since the lattice parameters did not undergo significant changes with doping, providing overall structural stability. The absence of the CuO phase may indicate that Cu^2+^ ions replaced Fe^3+^ ions at site A, which was confirmed by Rietveld refinement with the respective occupancy fractions 4, 8.5 and 14% (x = 0.05, 0.10, and 0.15). In addition, the majority phase of m-FeNbO_4_ was quantified as 96, 97, 98, and 100% for x = 0, 0.05, 0.10, and 0.15, respectively.

### 3.2. Raman Spectroscopy

According to group theory, the calculations for the space group P2/c (No. 13), with point symmetry C_2h_, showed that the 3N degrees of freedom for the 12 atoms in each primitive cell were divided into 36 modes of vibrations, as follows:Г_vibration_ = 8Ag + 10Bg +8Au + 10Bu,(1)
where all even vibrations (g) are Raman active and all odd vibrations (u) are IR active [26,27,28,29]. Of the 18 possible active Raman vibration modes, at least 16 distinct vibration bands were identified in the spectra shown in Figure 2a. Prominent vibration frequencies were located at 110, 135, 150, 175, 213, 226, 274, 303, 321, 362, 392, 416, 467, 498, 597, and 818 cm^−1^. The 622 cm^−1^ band observed for x = 0 was attributed to the vibration modes associated with the secondary phases of NbO_2_ and Nb_2_O_5_ [30], previously confirmed by XRD (Figure 1b). Based on the calculations of structures belonging to the same point group, it was possible to identify the attributions of each vibration mode [27,31] (see Table 2).

All 16 bands detected were attributed to the vibrations of the FeNbO_4_ monoclinic phase base units [4,8,32]. Low-frequency modes below 303 cm^−1^ are vibration modes that originate from network vibrations [32]. The most intense peaks identified in the regions of higher frequencies, 362, 597, and 818 cm^−1^, were associated with the frequencies of the internal vibration modes of the NbO_6_ octahedrons corresponding to the balance symmetries and stretching of the Nb-O bonds [32].

The strong interactions between ions caused sharper and more intense Raman modes [33]. Therefore, the most intense modes at 818, 597, and 362 cm^−1^ suggested the existence of strong interactions in the NbO_6_ octahedron.

The Raman spectra were fitted using the peak profile of the Lorentz distribution to confirm the effect of Cu^2+^ substitution on the vibrational characteristics of the Nb-O bonds; the results are shown in Figure 2c. The broad peak near 600 cm^−1^ was indicative of overlapping vibration modes, so we set it to three modes. The mode at 597 cm^−1^ was attributed to the m-FeNbO_4_ phase, that at 615 cm^−1^ was associated with the contribution of oxygen vacancies (Vo) [34], and that at 662 cm^−1^ corresponded with the vibration modes of Nb_2_O_5_ impurities. The Raman shift, width at half height, and the intensity of the Nb-O symmetric stretching modes at 818 cm^−1^ are shown in Figure 2d.

The increase in the Raman shift due to the rise in doping was attributed to the shrinkage of the Nb-O bonds [4]. The reduction in the NbO_6_ octahedron could be attributed to the insertion of ions with a smaller ionic radius in the A site, resulting in a smaller bond length (see Table 1). Additionally, a decrease in intensity and an increase in FWHM values were observed in the main vibration modes at 362, 597, and 818 cm^−1^, which implied that the degradation of the ordered structure and the disordered arrangement of the A-site ions were the causes of the Raman spectrum broadening [33]. Thus, the increase in the substitution of Fe^3+^ ions by Cu^2+^ions in the ordered structure increased the cationic disorder, which was directly reflected in the magnetic ordering of the structure.

### 3.3. Magnetic Properties

The compound FeNbO_4_, in the monoclinic phase, is reported in the literature to have a structure with a liquid antiferromagnetic order due to the FM and AFM competitions associated with the interactions between the octahedral Fe-O-Fe chains and the crossed Fe-O-chains. Nb-O-Fe [1,14,15]. The results obtained through magnetic characterizations, the magnetization measurements as a function of temperature (FC and ZFC), and the field-dependent magnetization are shown in Figure 3a, which confirmed the antiferromagnetic nature with a Néel temperature of approximately 36.81K. The magnetic transition temperature (TN) was defined by the derivative of the product of susceptibility with temperature (d(χT)/dT) and the derivative of susceptibility concerning temperature (dχ/dT), see Figure 3b. Generally, a peak in dχ/dT represents a PM (paramagnetic) to FM (ferromagnetic) transition region. For PM to AFM transitions, it has been shown theoretically and experimentally that the peak accurately determines TN in d(χT)/dT since χT is proportional to the magnetic energy near TN in an antiferromagnet. Thus, the peak in d(χT)/dT corresponds to the peak in specific heat [35]. In our case, the peaks of both derivatives were in the same region (36.81K), confirming the magnetic transition region.

The ZFC and FC curves showed similar behavior, with a smooth drop below TN, which may have been associated with weak AFM interactions in the structure [15] (Figure 3a). We also observed an increase in magnetization below 29K, which could be attributed to short-range spin interactions with a small fraction of Fe^3+^ in the niobium chains [15], as well as the contribution of α-Fe_2_O_3_ secondary-phase ferromagnetic interactions. Figure 3c provides the M × H curves measured for x = 0 at 5 and 300K, respectively. In the insert of Figure 3c, one can observe a small gap in the hysteresis curve for the M vs. H curve measured at 300K, with respective coercive field (Hc) and remanent magnetization (Mr) values of Hc = 201.5 Oe and Mr = 0.156 emu/g, which could be attributed to the contributions of the impurities present in the sample.

The inverse magnetic susceptibility data, obtained from the curve in ZFC mode, were fitted, as shown in the inset of Figure 3a. Above 60K, they displayed paramagnetic behavior according to the Curie–Weiss law (χ = C/(T − θ)), where C is the Curie–Weiss constant, θ_cw_ is the Curie–Weiss temperature, and T is the absolute temperature. The values obtained were C = 2.56 emu·K/g Oe and θ_cw_ = −122 K, respectively. The negative θ_cw_ value confirmed the antiferromagnetic nature. The effective magnetic moment was calculated using the Curie constant according tothe following relationship:(2)$$\mu_{\mathrm{eff}}^{\exp}=\sqrt{8C}\mu_{B},\;$$
where $\mu_{B}$ is the Bohr magneton [15]. The effective magnetic moment obtained was 4.53 $\mu_{B}$ for Fe^3+^, which agreed with the values reported in previous works [11,15].

The influence of Cu^2+^ insertion on the magnetic properties of FeNbO_4_ was evaluated as follows. Figure 3d shows the magnetization versus temperature curves (M–H) measured at 5 and 300K for x = 0.05. The insertion of 5% Cu^2+^ promoted a significant increase in magnetization and changed the behavior of the curves for low fields (see the inset in Figure 3d). This result indicated that the change in magnetic ordering in the structure began as a result of replacement. Figure 4a shows the M–H curves as a function of the increase in Cu^2+^ insertion, where one can observe a significant change in the curve’s behavior as a result of substitution, i.e., the magnetic transition induced by the insertion of Cu^2+^ ions into the FeNbO_4_ structure. The curves for x = 0.10 and 0.15 showed weak ferromagnetic behavior, with respective values of Hc = 85 Oe, Mr = 0.25 emu/g and Hc = 41 Oe, Mr = 0.19 emu/g (inset in Figure 4a). The M–H curves measured at 5K are shown in Figure 4b, where an increase in magnetization and the coercive field and remanent magnetization values can be observed: Hc = 140 Oe, Mr = 0.54emu/g and Hc = 73.3 Oe, Mr = 0.39 emu/g for x = 0.10 and 0.15, respectively (inset in Figure 4b).

To try to understand the main mechanisms responsible for the origin of magnetism in the doped FeNbO_4_ structure, the bound magnetic polaron (BMP) model was proposed:(3)$$M={\;M}_{0}L\left(x\right)+{\;\chi }_{m}H,$$
where the first term represents the contribution of the BMP, and the second term represents the contribution of the paramagnetic matrix. The value of M_0_ is equal to N_p_xm_s_, where N_p_ is the density number of BMPs involved per cm^3^, and m_s_ is the effective spontaneous magnetic moment per BMP. L(x) is the Langevin function defined as L(x) = coth(x)−1/x, where x = μ_eff_H/k_B_T. μ_eff_ is the true spontaneous moment per BMP at room temperature approximated by μ_eff_~m_s_ [36]. μ_eff_ can be expressed in Bohr magneton units, see Table 3. Figure 4c shows the fit of the M–H curves for the BMP model, where one can see that the experimental data perfectly fit the proposed model, and the theoretical parameters extracted can be seen in Table 3.

The theoretical BMP model affirmed that the presence of magnetic carriers and defects such as oxygen vacancies were the main factors responsible for the magnetism of defects induced by carriers, which formed magnetic polarons. Therefore, substitution with a lower oxidation state could increase the generation of oxygen vacancies that behave like polarons, altering the material’s magnetic properties [36].

The increased insertion of Cu^2+^ ions caused a magnetic transition from antiferromagnetic to weak ferromagnetic. The increase in the initial saturation magnetization as a result of substitution can be seen in Figure 4d. Our results showed that the origin of magnetism in the Fe_1−x_Cu_x_NbO_4_ may have been associated with the contribution of super-exchange interactions between Cu^2+^-O-Fe^3+^ as well as induced defects such as oxygen vacancies that behaved as bound magnetic polarons. Figure 5 shows an illustrative scheme that helps elucidate the change in magnetic ordering and the contribution of each interaction. Figure 5a shows a net AFM configuration in the structure as already reported in the literature for m-FeNbO_4_ [10]. This configuration contains FM interactions associated with Fe-O-Fe interactions and AFM interactions associated with Fe chains mediated by Nb, resulting in a liquid AFM order. In Figure 5b, an FM configuration for x = 0.10 is provided as an example. The yellow underlined circle shows a possible FM interaction for Cu^2+^-O-Nb-O-Fe^3+^. The theoretical magnetic moment of copper (μ_Cu_ = 1.73μ_B_) is much smaller than that of iron (μ_Fe_ = 5.9μ_B_), so the substitution could promote this configuration state. However, these interactions alone may not be enough to encourage the transition or explain our results. In this way, the polaron model was proposed and helped us to understand the contribution of defects to the magnetic properties of our samples.

In Figure 5b, the area highlighted in red shows the regions of possible magnetic polaron formation due to oxygen vacancies. The center of a polaron is occupied by a defect, such as an oxygen vacancy, which would be responsible for aligning the spins of the closest elements. Atoms that are not close enough to an oxygen vacancy do not align and may even form an AFM configuration [37].

The fit data can show us the density of polarons (Np) formed, which was in the order of 10^20^–10^23^ and increased as more polarons formed (see Table 3); this order of magnitude was sufficient to make significant contributions to the magnetic properties [38]. Each polaron formed could contribute an effective polaron magnetic moment (μ_p_) of approximately 1.2 μ_B_. Thus, we could determine that the AFM/FM transition was the origin of FM interactions between Cu^2+^-O-Fe^3+^ ions, with a contribution from defects.

## 4. Conclusions

We prepared samples of Fe_1−x_Cu_x_NbO_4_ (x = 0∓0.15) by the conventional solid-state reaction method. Structure analysis by XRD and Raman spectroscopy confirmed the monoclinic phase of FeNbO_4_ and the presence of impurities (Fe_2_O_3_, NbO_2_, and Nb_2_O_5_). The refinement experiments confirmed the substitution of Cu^2+^ in the Fe site and quantified the obtained phase fractions. The increase in the insertion of Cu^2+^ did not cause significant changes in the network parameters. However, it promoted local changes in each network site. The Raman spectroscopy results complemented and confirmed the XRD results. The results confirmed the monoclinic phase and the local changes promoted by the insertion of Cu^2+^ (oxygen vacancies). Studies of the magnetic properties confirmed the AFM nature of the pure sample, as well as evidence of a magnetic transition caused by the insertion of Cu^2+^ into the structure. The insertion of ions in the lower oxidation state induced the generation of oxygen vacancies that behaved like magnetic polarons and contributed to the change in magnetic ordering. Adjustments to the bound magnetic polaron model confirmed the contribution of structural defects to the magnetic properties. The AFM to FM transition could be associated with exchange interactions between Cu^2+^-O-Fe^3+^ ions and the contribution of defects such as oxygen vacancies.

## Figures and Tables

**Figure 1 materials-15-07424-f001:**
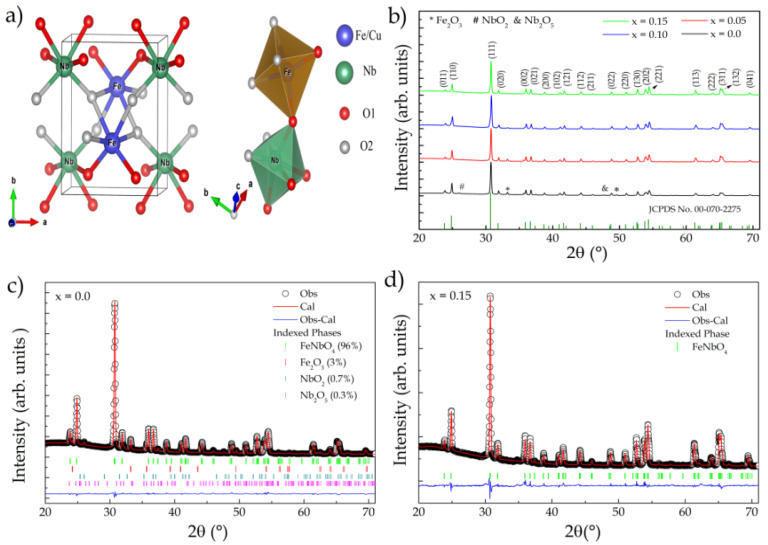
(**a**) Structural representation of FeNbO_4_; (**b**) XRD standard for Cu_x_Fe_1−x_NbO_4_ samples (x = 0–0.15); (**c**,**d**) adjustment profiles for x = 0 and 0.15, respectively (Rietveld refinement).

**Figure 2 materials-15-07424-f002:**
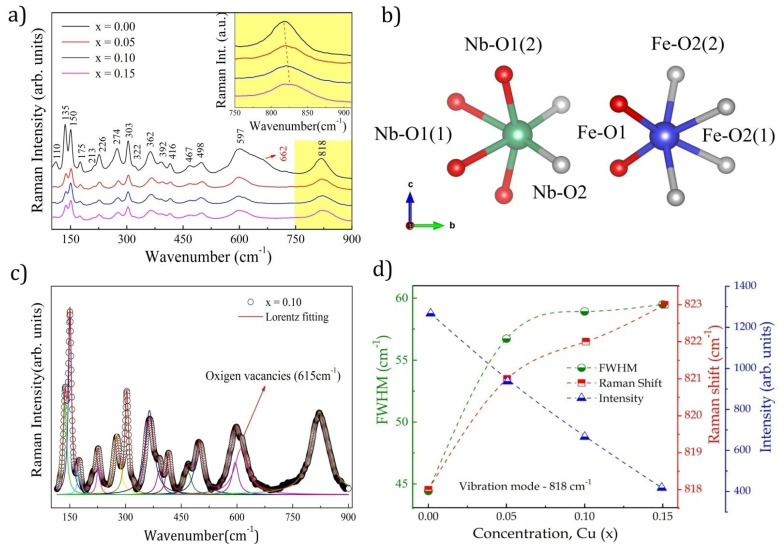
Raman Fe_1−x_Cu_x_NbO_4_ spectroscopy (x = 0.0-0.15). (**a**) inset shows the peak magnification at 818 cm^−1^ evidencing the Raman shift as a function of the Cu insertion; (**b**) schematic representation of NbO_6_ and FeO_6_ octahedral symmetry and their respective connections; (**c**) fit profile calculated for x = 0.10, the colored lines show the deconvolution of peaks; (**d**) FWHM, intensity, and Raman shift due to Cu substitution.

**Figure 3 materials-15-07424-f003:**
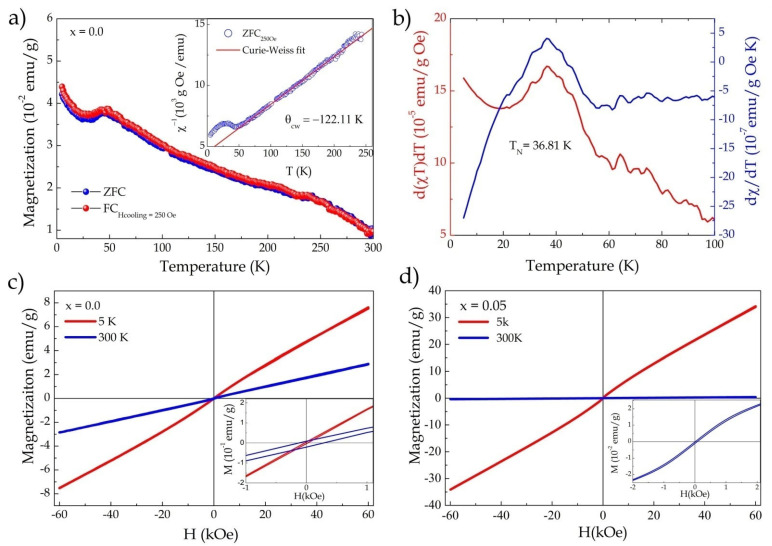
(**a**) Magnetization as a function of temperature (FC and ZFC) with an applied field of 250 Oe (FC)—inset shows the inverse of susceptibility as a function of temperature (χ^−1^/T) adjusted by the Curie–Weiss law; (**b**) derivative of the product of susceptibility with temperature (d(χT)/dT) and derivative of magnetic susceptibility (dχ/dT) as a function of temperature for x = 0.0, showing the Néel transition temperature; (**c**) magnetization curves as a function of applied field (M–H) measured at 5 and 300 K for x = 0.0; (**d**) M–H curves measured at 5 and 300 K for x = 0.05.

**Figure 4 materials-15-07424-f004:**
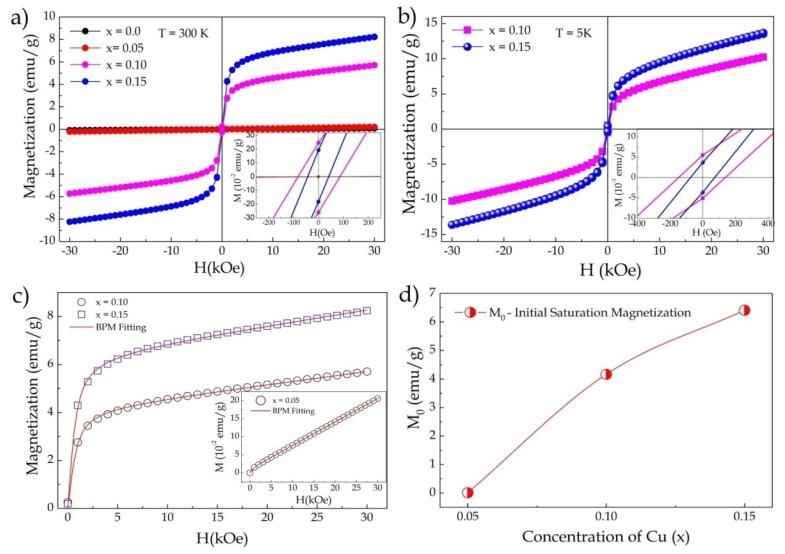
(**a**) Hysteresis curves (M–H) at room temperature for x = 0–0.15; (**b**) M–H curves measured at 5 K for x = 0.10 and 0.15; (**c**) adjustments of the M–H curves at 300K for the BMP model for x = 0.05, 0.10, and 0.15; (**d**) initial saturation magnetization (M_0_) as a function of Cu^2+^ substitution extracted from the fitted model.

**Figure 5 materials-15-07424-f005:**
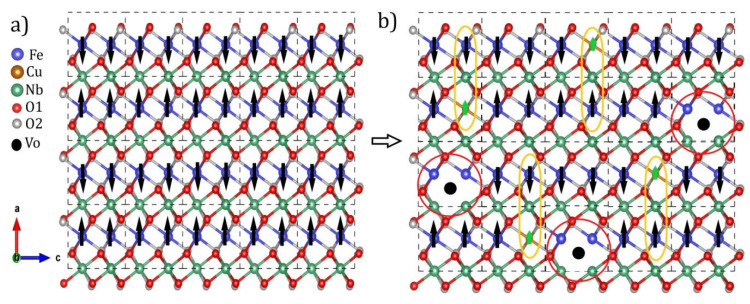
Representative illustration of the magnetic ordering configurations: (**a**) AFM and (**b**) FM.

**Table 1 materials-15-07424-t001:** Structural parameters and quality factors obtained by refinement.

**Lattice parameters**	**Samples**	**0.00**	**0.05**	**0.10**	**0.15**
	a (Å)	4.650(3)	4.651(6)	4.651(5)	4.652(9)
	b (Å)	5.616(8)	5.619(0)	5.620(2)	5.623(7)
	c (Å)	4.996(3)	4.99(0)	4.997(5)	4.999(5)
	α(⁰)	90	90	90	90
	β(⁰)	90.181(4)	90.130(4)	90.120(9)	90.079(4)
	γ(⁰)	90	90	90	90
	V(Å^3^)	130.5	130.42	130.64	130.81
**Bond length**	Fe-O1 × 2(Å)	1.93	1.95	2.03	1.9
	Fe-O2 × 2(Å)	2.04	2.12	2.18	2.1
	Fe-O2 × 2(Å)	2.07	2.02	2.04	2.11
	Nb-O1 × 2(Å)	2.02	1.87	1.96	1.84
	Nb-O1 × 2(Å)	2.2	2.16	2.05	1.98
	Nb-O2 × 2(Å)	1.86	1.87	1.83	1.84
**Bond angle**	Fe-O-Fe(⁰)	98.84	99.09	97.5	96.94
	Nb-O-Fe(⁰)	125.19	127.95	123.43	129.03
**Quality factors**	χ^2^	7.034	2.61	6.95	4.50
	R_wp_ (%)	3.21	2.29	3.82	3.13
	R_p_ (%)	2.20	5.5	3.13	8.00

**Table 2 materials-15-07424-t002:** Active Raman modes and assignments.

Band No.	Symmetry	ω_obs_ (cm^−1^)				Assignments
		0	0.05	0.10	0.15	
1	Bg	110				Rotational mode of NbO_6_ around an axis perpendicular to the b-axis
2	Ag	135	137	137	137	Rotational mode of ${\mathrm{NbO}}_{4}^{3-}$ around the b-axis
3	Bg	150	151	150	150	Rotational mode of ${\mathrm{NbO}}_{4}^{3-}$ around an axis perpendicular to the b-axis
4	Bg	175	176	175	175	Translational mode parallel to the ac-plane
5	Ag	213	212	212	212	Translational mode along the b-axis
6	Bg	226	227	226	225	Translational mode parallel to the ac-plane
7	Ag	274	277	276	276	Translational mode along the b-axis
8	Bg	303	304	303	303	Translational mode parallel to the ac-plane
9	Ag	322	322	321	321	Scissors mode of ${\mathrm{NbO}}_{4}^{3-}$
10	Ag	362	366	364	364	Scissors mode of ${\mathrm{NbO}}_{4}^{3-}$
11	Bg	392	394	391	392	Translational mode parallel to the ac-plane
12	Bg	416	415	415	415	Rocking mode of ${\mathrm{NbO}}_{4}^{3-}$
13	Bg	467	468	468	468	Rocking mode of ${\mathrm{NbO}}_{4}^{3-}$
14	Ag	498	500	498	500	Twist mode of ${\;\mathrm{NbO}}_{4}^{3-}$ around the b-axis
15	Bg	597	598	597	601	Stretching mode of ${\mathrm{NbO}}_{4}^{3-}$
16	Ag	818	821	822	823	Stretching mode of ${\mathrm{NbO}}_{4}^{3-}$

**Table 3 materials-15-07424-t003:** Parameters obtained by fitting to the BMP model.

Samples (x)	M_0_ (emu/g)	χ_m_ (emu/g Oe)	m_s_ (emu)	N_p_ (cm^−3^)	μ_eff_ (μB)
0.05	0.01139 (±0.00005)	0.00204 (±0.00008)	1.32 × 10^−23^	8.62 × 10^20^	1.42
0.10	4.166 (±0.03)	0.0196 (±0.0009)	1.11 × 10^−23^	3.74 × 10^23^	1.20
0.15	6.40 (±0.05)	0.0233 (±0.001)	1.14 × 10^−23^	5.61 × 10^23^	1.23

## Data Availability

Not applicable.

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
