# Peer review of "Antiferromagnet–Ferromagnet Transition in Fe1−xCuxNbO4"

_materials, 2022, doi:10.3390/ma15217424_

Round 1
Reviewer 2 Report
This paper describes well on the synthesis and characterization by x-ray diffraction, Raman spectroscopy, and magnetic measurements of Fe1-xCuxNbO4 (samples with x = 0-0.15) prepared by the modified solid-state reaction method. The manuscript discusses work that will be of interest to the wider community and should be considered for publication in Materials. The work is interesting and written correctly, the authors should only pay attention to the numbering of the table - in line 242, table 1 appears instead of table 3. In Figure 3, the y-axis values should be separated by a dot instead of a comma.
Reviewer 3 Report
The authors present the AFM-FM transition of iron niobates by subsisting iron with copper. The results are interesting, which can be published in Materials with a proper revision with the following issues incorperated:
1. The figure caption of Fig.3 is incorrect. For example, I can only see the results for 300 K in Fig.3 (a) while the caption claimed both 5 K and 300 K. There are similar mistakes for Fig.3 (d), Fig. 4 (a) and 4 (c).
2. More details should be given for Eq. (2), including the reference, the physical meaning and the unit of c. It should be noted that if c is not dimensionless, the unit of the left and right side of Eq. (2) is not the same.
3. More details should be given for Eq. (3). It is confused whether it is called MPML (Line 228), or rather MPM (line 230). Please give the physical meaning of all parameters.
4. Line 240, Tab 3 is mentioned. However, I cannot see where Tab 3 is.
Round 2
Reviewer 1 Report
The authors answered all the questions which were raised and implemented all the changes which were suggested. I believe the manuscript "materials-1965078" has been sufficiently improved to publish in Materials.